# Trend of Changes in Chloramphenicol Resistance during the Years 2017–2020: A Retrospective Report from Israel

**DOI:** 10.3390/antibiotics12020196

**Published:** 2023-01-18

**Authors:** Hannan Rohana, Anat Hager-Cohen, Maya Azrad, Avi Peretz

**Affiliations:** 1The Clinical Microbiology Laboratory, The Baruch Padeh Medical Center, Poriya, Tiberias 1528001, Israel; 2Azrieli Faculty of Medicine, Bar-Ilan University, Safed 1311502, Israel

**Keywords:** chloramphenicol, antibiotic resistance, Israel, trends of changes

## Abstract

Resistant bacteria limit treatment options. This challenge has awakened interest in antibiotics that are no longer in use due to side effects, such as chloramphenicol. This work investigated trends in chloramphenicol resistance rates during 2017–2020 in bacteria isolated from diverse clinical samples at the Baruch Padeh Medical Center, Poriya, Israel. Bacteria were isolated from 3873 samples and identified using routine methods, including matrix-assisted laser desorption ionization-time of flight (MALDI-TOF) technology. Chloramphenicol susceptibility was tested using a VITEK II instrument or by the Kirby–Bauer disk-diffusion test. The average chloramphenicol resistance rate was 24%, with no significant differences between study years. Chloramphenicol resistance was associated with sample origin (*p* < 0.001); isolates originating from sputum samples showed 49.8% resistance rate, compared to 2.3% of the body fluid isolates, 10.4% of the ear/eye isolates and 22.5% of the blood isolates. Furthermore, there was a significant decrease in chloramphenicol resistance among blood and ear/eye isolates during the study period (*p* = 0.01, *p* < 0.001, respectively). The highest resistance rate was among *Pseudomonas aeruginosa* isolates (50.5%). In conclusion, since chloramphenicol susceptibility seems to be retained, its comeback to the clinical world should be considered.

## 1. Introduction

Despite considerable advances in the management of infectious diseases, the frequent prescription of antibiotics by healthcare providers worldwide has led various pathogens to develop considerable resistance patterns. The rising resistance rates to primary antimicrobial agents, alongside the limited treatment options, has revived interest in old antibiotics that are no longer in use [1,2]. One such antibiotic is chloramphenicol (CA), which was initially found in *Streptomyces* spp. isolated from a soil sample [3]. In 1949, it was approved for clinical use in the United States, but given its association with aplastic anemia and dose-related bone marrow suppression, physicians tend to use it cautiously. Nonetheless, in developing countries, it is still considered a first-line treatment for many infections such as enteric fever [4].

CA exhibits broad-spectrum activity against both Gram-negative and Gram-positive bacteria [2]. Generally, it is considered a bacteriostatic antibiotic, which inhibits protein synthesis, by binding to the 50S ribosomal subunit, inhibiting the formation of peptide bonds between amino acids, which subsequently limits bacterial growth [5]. It is also bactericidal at high concentrations or against specific bacteria, such as *Neisseria meningitides*, *Haemophilus influenzae* and *Streptococcus pneumoniae* [2,6]. Among other advantages of this antibiotic are its excellent tissue penetration, low cost and effectiveness both in parenteral and oral forms. In addition, therapeutic levels can be achieved in pleural, ascitic, synovial and cerebrospinal fluids [2,7]. 

Typhoid was one of the first infections treated with CA [8]. Other indications for CA are anaerobic infections, bacterial meningitis in penicillin-allergic patients, rickettsial infections and brain abscesses [9]. It was also used in veterinary [8]. Despite its many benefits, CA has fallen into desuetude due to reported adverse side effects [2,7,8]. For example, aplastic anemia, a condition in which insufficient white and red blood cells are produced by the bone marrow, has been associated with CA use [8]. However, the overall incidence of aplastic anemia following CA use was estimated to be less than one case per 10 million per year. Reports of apparent genotoxic carcinogenicity of CA were in essence the trigger to the current practice to use it sparingly. Nevertheless, only limited data regarding this risk exist, considering that CA use has almost ceased and no new studies are performed in relation to CA carcinogenicity [8]. Yet, many developing countries still use CA for treatment of rickettsial infections, bacterial meningitis in patients allergic to penicillin, anaerobic infections and typhoid fever [7]. 

Resistance to CA has developed through several mechanisms, including enzymatic inactivation by acetylation via different types of chloramphenicol acetyltransferases, target site mutations, phosphotransferase inactivation, permeability barriers and efflux systems [9,10,11]. However, in recent years, a trend toward increased susceptibility has been noted, apparently due to its limited use [7]. This study aimed to investigate trends of CA resistance in recent years, among bacteria isolated from diverse clinical samples. This is a successor study to a previous study conducted in 2015 in Israel to characterize CA susceptibility patterns of pathogens [7]. 

## 2. Results

The study included 3873 isolates; 1250 (32.3%) isolates originated from blood, 1396 (36.0%) from ear/eye, 217 (5.6%) from fluids and 1010 (26.1%) from sputum samples. The isolates belonged to the following bacterial families: *Enterobacteriaceae* (37.3%), *Staphylococcaceae* (26%), *Pseudomonadaceae* (16.4%), *Streptococcaceae* (8.7%), *Enterococcaceae* (3.8%), *Moraxellaceae* (3.8%), *Yersiniaceae* (2%), *Xanthomonadaceae* (1%), *Alcaligenaceae* (0.3%), *Aeromonadaceae* (0.2%), *Corynebacteriaceae* (0.2%), *Pasteurellaceae* (0.2%) and *Bacillaceae* (0.1%).

The resistance rate to CA was 26% in 2017, 25.5% in 2018, 21.9% in 2019 and 22.4% in 2020. The four-year average resistance rate was 24% (Table 1, Figure 1). 

A significant association was found between sample origin and CA resistance (*p* < 0.001). More specifically, 49.8% of the isolates originating from sputum samples were resistant, compared to 2.3% of the body fluids, 10.4% ear/eye and 22.5% blood isolates (Table 2).

Comparison of CA resistance by both year and sample origin found a significant decrease in CA resistance among isolates from blood (*p* = 0.01) and ear/eye (*p* < 0.001) samples between 2017 and 2020 (Table 3). 

Analysis of CA resistance rates among the different bacterial species found significant differences in resistance rates (*p* <0.001) (Table 4), with the highest resistance rate among *Pseudomonas aeruginosa* (*Ps. Aeruginosa*) isolates (50.5%), followed by *Coagulase-negative staphylococci* (CNS) (27.3%). Resistance rates among samples carrying *Klebsiella pneumoniae* (*K. pneumonia*)*, Escherichia coli* (*E. coli)* and *Staphylococcus aureus* (*S. aureus*) were 19.8%, 10.1% and 15.6%, respectively. 

Moreover, a significant association was found between CA resistance and bacteria type in blood and ear/eye samples (*p* < 0.001). According to a post hoc Chi-squared test, the highest resistance rate in blood isolates was noted among samples carrying CNS (38.7%), followed by *S. aureus* and *K. pneumoniae* (20% and 19.1%, respectively). The highest resistance rate in ear/eye isolates was seen in samples carrying CNS (19.4%) or *K. pneumoniae* (17.9%).

## 3. Discussion

Given the increasing bacterial resistance to a wide variety of antibiotics, we investigated the resistance rates of different bacterial isolates to CA, in order to assess its potential to be reintroduced into treatment protocols, especially in cases of multi-drug resistant pathogens. In particular, we wanted to explore the resistance rates of *Enterobacteriaceae*, *Staphylococcaceae* and *Ps. aeruginosa* to CA among various clinical samples.

Overall, there were no significant changes in CA resistance rates during 2017–2020. However, when focusing on sample origin, CA resistance was found to decrease significantly in isolates from ear/eye and blood samples during these years. This can be explained by the limited use of CA; although CA’s usage covers a broad spectrum of bacteria and is quite inexpensive, its use has been generally limited due to its possible association with aplastic anemia as mentioned above [12]. Given the fact that *S. aureus* and CNS (both *Staphylococci*) are the main etiological agents of eye infections, first-treatment with CA has been replaced by fluoroquinolones, as they are highly efficacious against both Gram-negative and Gram-positive bacteria, particularly *Staphylococci* [13,14]. However, it should be noted that the emerging resistance to fluoroquinolones in recent years, might accelerate the comeback of CA.

In the current study, isolates from sputum samples presented the highest resistance rate (49.8%) to CA. Back in 2012, Nitzan et el. conducted a national survey that included 18 Israeli hospitals, to evaluate CA use and to assess bacterial susceptibility to CA [7]. Eight (44.4%) out of 18 hospital microbiology laboratories routinely assessed CA susceptibility of different isolates. The resistance rate among sputum samples was 25% [7], almost half the resistance rate in our research. This suggests that CA use for treatment of respiratory tract infections increased in recent years in Israel. In contrast, a study conducted in Ethiopia reported that 84.6% of the *S. aureus* isolated from nasal colonization, were sensitive to CA [15]. Thus, further studies should be performed to determine whether CA efficacy differs in the various body compartments.

Among the different bacterial species investigated here, *Ps. aeruginosa* isolates demonstrated the highest resistance rate to CA (50.5%). This was not surprising, as most strains of *Ps. aeruginosa* present intrinsic resistance to a wide variety of antibiotics, including CA [16]. This may be due to its outer membrane, which exhibits very low nonspecific permeability to small hydrophilic molecules [17]. Moreover, to date, seven resistance-nodulation-division (RND) multidrug efflux systems have been described [18]. These systems are likely to contribute significantly to the intrinsic resistance of *Ps. aeruginosa* to different antimicrobial agents, including CA [16].

Comparison of CA resistance rates among the different bacterial species by sample origin found the highest resistance rate in blood CNS isolates (38.7%). In a study conducted in Nepal, which assessed CA resistance among CNS samples isolated from blood cultures, only 8.9% of the isolates were resistant to CA [19]. Such a low rate of CA resistance is consistent with another study [20]. This disparity may result from distinctly different use patterns of CA across countries. Nevertheless, the high CA resistance rate in our study requires further investigation and might be attributed to injudicious usage of CA among patients suffering from sepsis.

Research on CA susceptibility over the past two decades has mostly been conducted in developing countries. Low resistance rates (13.6%) were noted among Gram-positive bloodstream isolates from Iran during 2001–2004 [21], and ten years later among methicillin-resistant *S. aureus* (26%) in developing countries, such as Brazil and Nigeria [22]. Thus, the use of CA is accompanied by a rise in resistance rate. Nevertheless, this resistance rate is still tolerable. Certainly, there is a need for performance of similar studies in developed countries in order to consider the use of CA. The average resistance rate of CA in our study was 24%, which indicates that it can be used safely for treatment of different bacterial infections.

## 4. Materials and Methods

### 4.1. Bacterial Isolates

This retrospective study included 3873 isolates of different bacterial families, including *Enterobacteriaceae*, *Streptococcaceae*, *Staphylococcaceae*, *Pseudomonadaceae* and *Moraxellaceae*. The bacterial isolates were all recovered from the microbiology laboratory at Baruch Padeh Medical Center, Poriya, from clinical samples of patients admitted to the medical center between 2017 and 2020.

All bacteria were isolated and identified using routine clinical microbiology laboratory methods, with final identification performed using matrix-assisted laser desorption ionization-time of flight (MALDI-TOF) technology (Bruker Daltonics, Bremen, Germany). This study focused on *Pseudomonas aeruginosa* (*Ps. Aeruginosa*, *N* = 628), *Coagulase-negative staphylococci* (CNS, *N* = 610), *Escherichia coli* (*E. coli*, *N* = 607), *Staphylococcus aureus* (*S. aureus*, *N* = 397) and *Klebsiella pneumoniae* (*K. pneumoniae*, *N* = 394).

The work was carried out according to protocols and guidelines of the institutional review board of Padeh Poriya Medical Center.

### 4.2. Antibiotic Susceptibility Testing

CA sensitivity was assessed using the VITEK II instrument (Bio-Merieux, Marcy-L’etoile, France) or the Kirby–Bauer disk-diffusion test, with results interpreted according to the Clinical Laboratory Standards Institute (CLSI) Performance Standards for Antimicrobial Susceptibility Testing guidelines.

### 4.3. Statistical Analysis

All categorical variables are presented as absolute numbers and percentages. Chi-squared tests were performed to assess the relationship between CA resistance and year, origin of sample and type of bacteria. A *p* value < 0.05 indicated statistical significance. Statistical analysis was performed using R Statistical Software (version 4.1; R Foundation for Statistical Computing, Vienna, Austria).

## 5. Conclusions

CA was one of the first antimicrobial agents discovered, and exhibits a broad antibacterial spectrum. According to our data, CA susceptibility has been retained, apparently due to its limited usage. Therefore, in the current era with high antibiotic resistance and with no new treatments, reintroduction of CA into standard antibacterial infection management protocols should be considered. Based on the current study’s results, clinicians can assess CA treatment benefits based on bacteria type and on sample origin. Further studies in other developed countries will be necessary to strengthen these findings and to assure that the adverse effects rate can be tolerated.

## Figures and Tables

**Figure 1 antibiotics-12-00196-f001:**
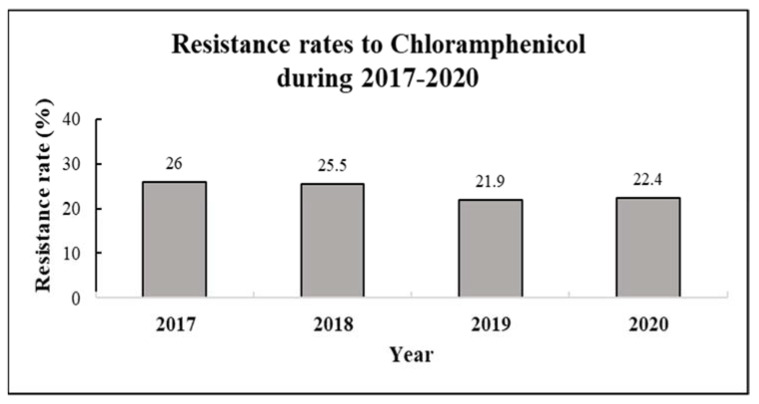
Resistance rates to chloramphenicol by year (2017–2020). Resistance rates were calculated for 3873 various clinical samples collected during 2017–2020, with the following distribution—*n* = 1005 (2017), *n* = 984 (2018), *n* = 1011 (2019, and *n* = 843 (2020).

**Table 1 antibiotics-12-00196-t001:** Chloramphenicol susceptibility patterns by year.

	2017 (*N* = 1005) *n* (%)	2018 (*N* = 984) *n* (%)	2019 (*N* = 1011) *n* (%)	2020 (*N* = 873) *n* (%)	Total (*N* = 3873) *n* (%)	*p* Value *
Resistant	261 (26)	250 (25.5)	220 (21.9)	185 (22.4)	916 (24)	0.070
Susceptible	741 (74)	732 (74.5)	785 (78.1)	641 (77.6)	2899 (76)	

* Chi-squared test.

**Table 2 antibiotics-12-00196-t002:** Chloramphenicol resistance rates by sample origin.

	Blood (*N* = 1250) *n* (%)	Ear/Eye (*N* = 1396) *n* (%)	Fluids (*N* = 217) *n* (%)	Sputum (*N* = 1010) *n* (%)	*p* Value *
Resistant	275 (22.5)	145 (10.4)	5 (2.3)	491 (49.8)	<0.001
Susceptible	946 (77.5)	1246 (89.6)	212 (97.7)	495 (50.2)	

* Chi-squared test.

**Table 3 antibiotics-12-00196-t003:** Chloramphenicol resistance rates by year and sample origin.

Sample’s Origin	2017 (*N* = 1005) *n* (%)	2018 (*N* = 984) *n* (%)	2019 (*N* = 1011) *n* (%)	2020 (*N* = 873) *n* (%)	*p* Value *
Blood	79 (27.6%)	83 (25.4%)	67 (19%)	46 (18%)	0.01
Ear/eye	45 (11.5%)	60 (14.8%)	24 (6.9%)	16 (6.5%)	<0.001
Sputum	134 (50%)	107 (53.2%)	128 (52.2%)	122 (44.9%)	0.241
Fluids	3 (5.2%)	0 (0%)	1 (1.8%)	1 (1.9%)	0.337

* Chi-squared test.

**Table 4 antibiotics-12-00196-t004:** Chloramphenicol resistance rates by bacteria type.

	*Ps. aeruginosa* (*N* = 628) *n* (%)	CNS * (*N* = 610) *n* (%)	*E. coli* (*N* = 607) *n* (%)	*S. aureus* (*N* = 397) *n* (%)	*K. pneumoniae* (*N* = 394) *n* (%)	*p* Value **
Resistant	317 (50.5)	166 (27.3)	59 (10.1)	62 (15.6)	78 (19.8)	<0.001
Susceptible	311 (49.5)	443 (72.7)	527 (89.9)	335 (84.4)	316 (80.2)	

* CNS = Coagulase negative *Staphylococci.* ** Chi-squared test.

## Data Availability

The dataset used and/or analyzed during the current study are available from the corresponding author on reasonable request.

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
