# Peer review of "Trend of Changes in Chloramphenicol Resistance during the Years 2017–2020: A Retrospective Report from Israel"

_antibiotics, 2023, doi:10.3390/antibiotics12020196_

Round 1

Reviewer 1 Report

In the method section, "This retrospective study included 3873 isolates of different bacterial families including Enterobacteriaceae, Streptococcaceae, Staphylococcaceae, Pseudomonadaceae and Moraxellaceae." is expressed. In the study, the distribution of other species and the resistance rate are given in the findings, while "Enterobacteriaceae, Streptococcaceae, and Moraxellaceae." no numbers and resistance ratios of the genus are given. It is thought that it would be useful to give the total number of samples and the species distribution.

Bacteria names have italic misspellings and these need to be checked.

Author Response

Dear reviewer

We thank you for your comments.

  • We added the distribution of pathogens according to families (Lines  71-76)
  • The manuscript was re-edited by a language editor.

Reviewer 2 Report

The article presents the resistance to chloramphenicol of various types of bacteria isolated from different sources. The study is original and important for further antibiotic therapies. Although is nicely presented I have some suggestions to make. Please include in your study the possible influence of the country. There are studies which show a high prevalence of resistance to Chloramphenicol in bacteria isolated from various sources (food bacteria, animal origin bacteria). The fact that in Israel the susceptibility is higher is interesting and should be taken into consideration when discussing the matter. It would be more interesting to add to this study a molecular analysis for CHL resistance gene presence in the isolated bacteria. Overall, the article is well written and debates an interesting subject. 

Author Response

Dear reviewer, 

We thank you for your comments.

Unfortunately, we have not find any reference for possible influence of the country.

Reviewer 3 Report

Introduction

Add information about the few use of CA

Check if Streptomyces spp or sp?

Review the spelling  of the phrase It is also bactericidal against Neisseria meningitides, Haemophiles influenzae, and Streptococcus pneumoniae or at high concentrations

The conclusion must be improved 

Author Response

Dear reviewer,

We thank you for your comments.

  • We added information about the use of CA (Lines 48-50).
  • We corrected Streptomyces to spp
  • We rephrased the sentence "It is also bactericidal against Neisseria meningitides, Haemophiles influenzae, and Streptococcus pneumoniae or at high concentrations: (Lines 42-44). Additionally, the manuscript was re-edited by a language editor. 
  • The conclusion section was revised (Lines 201-208).